# Tryptase: The Silent Witness of Past and Ongoing Systemic Events

**DOI:** 10.3390/medicina60091380

**Published:** 2024-08-23

**Authors:** Irena Oštrić Pavlović, Sara Radović, Danka Krtinić, Jelena Spirić, Nataša Kusić, Antonije Veličković, Vesna Tomić-Spirić

**Affiliations:** 1Clinic of Allergology and Immunology, University Clinical Centre of Serbia, 11000 Belgrade, Serbia; sara.radovic21@gmail.com (S.R.); kusic.natasa@gmail.com (N.K.); antonije.velickovic@gmail.com (A.V.); vesnatomicspiric63@gmail.com (V.T.-S.); 2Faculty of Medicine, University of Belgrade, 11000 Belgrade, Serbia; dd.krtinic@gmail.com; 3Clinic of Gastroenterology and Hepatology, University Clinical Centre of Serbia, 11000 Belgrade, Serbia; spiricjelena77@gmail.com

**Keywords:** tryptase, mast cell, anaphylaxis, hypersensitivity

## Abstract

*Introduction:* Tryptase is an important biomarker widely used in the laboratory confirmation of severe hypersensitivity reactions, especially anaphylaxis. It also plays a crucial role in the diagnosis, risk stratification, management and prognostic evaluation of many other mast cell-related conditions. *Aim:* This paper aims to highlight the role of serum tryptase, both in allergic disorders and other mast cell-related conditions. Two clinical cases regarding timely serum tryptase acquisition (in drug hypersensitivity reactions during the imaging procedure and perioperative anaphylaxis) are meant to emphasize the clinical potential of this protease. *Method:* We performed a comprehensive literature search of the PubMed/Medline and Scopus databases. From a total of 640 subject related publications, dating from 1940 to 2024, 45 articles written in English were selected. *Literature search results:* Total serum tryptase is a simple, cost-effective analysis with a normal baseline tryptase (sBT) level below 8.4 µg/L. Elevated sBT can indicate hereditary alpha-tryptasemia (HαT), mastocytosis and other non-allergic disorders. Patients with higher sBT levels, especially with insect venom allergy, have an increased risk of severe reactions and thereby require a prolonged treatment. All immediate systemic hypersensitivity reactions require a correlation between serum acute tryptase (sAT) and sBT. According to the guidelines, measuring sAT 30 min to 2 h after the symptom onset and sBT 24 h after the resolution, using the 20 + 2 rule and an sAT/sBT ratio of 1.685, improves the diagnostic accuracy in anaphylaxis. *Conclusions:* Tryptase levels should be acquired in all cases with clinical suspicion of MC degranulation. Given the increasing clinical relevance, elevated baseline serum tryptase levels require a multidisciplinary approach and further investigation.

## 1. Introduction

Since the discovery of the protease within the cells that play pivotal roles in immunological and hematological cascades, tryptase has drawn attention as a versatile, easily obtainable and reliable biomarker. Since the 1980s, scientific interest has primarily focused on allergic and genetic mastocyte-related disorders, particularly mastocytosis. However, more recently, tryptase has emerged as a valuable diagnostic and prognostic laboratory indicator in a variety of clinical conditions [1].

Despite vast amounts of data covering the biochemical characteristics of tryptase and its relevance in scientific research, the clinical implications for its use in everyday practice remain scarce.

The objective of this paper is to provide a thorough insight into tryptase biology and its role in the pathophysiology of hypersensitivity reactions and mast cell-related conditions, while also highlighting its growing number of clinical implications (represented in Table 1).

## 2. Method

We performed a comprehensive literature search of the PubMed/Medline and Scopus databases up to 1 May 2024, using the following search terms: tryptase, mast cell-related conditions, anaphylaxis, hypersensitivity and drug allergy.

Since 1940, over 640 papers have been dedicated to evaluating the various clinical roles and practical uses of tryptase. Following the detailed analysis of the available review papers and case reports, 45 articles were picked by the authors. Although there were no restrictions regarding the publication date, the majority of the selected papers were found to be contemporary, comprehensible, clinically relevant to date and have been published in the past twenty years. All of the articles have been published in English. Two clinical cases presented in this paper involved patients that were treated at the Clinic of Allergology and Immunology of the University Clinical Center of Serbia during the past five years.

## 3. Serum Tryptase: Biology and Pathophysiology

Mast cells (MCs) are long-lived, tissue-resident cells that play important roles in hypersensitivity and allergic reactions, as well as inflammatory responses. They are bone marrow-derived from hematopoietic lineage, originating from the CD34+/CD117+pluripotent progenitor cells [1]. Their maturation and differentiation take place throughout the body, under the influence of the c-KIT ligand stem cell factors, along with the growth factors provided by the microenvironment of the peripheral tissue where MCs are destined to reside [2]. Although mature MCs can be collocated to the blood vessels of any epithelial or mucosal tissue, most MCs are found at the crossroads between the host and the foreign antigen (skin, respiratory, gastrointestinal tract and ocular tissues).

Mature MCs contain numerous cytoplasmic granules that store pro-inflammatory mediators, including histamine, proteoglycans like heparin and many neutral serine proteases such as tryptase. Upon mast cell activation and degranulation, the contents of these granules are released, leading to tissue damage during immediate hypersensitivity reactions [1,2].

Although basophiles contain similar granular content, they have approximately 500 times less tryptase than mast cells. Consequently, elevated serum tryptase levels are considered to be a specific indicator of mast cell activation, leading to it being commonly referred to as mast cell-derived tryptase.

Tryptase is a serine esterase with four non-covalently bound subunits. There are four known types of human tryptase: α, β, γ and δ, but only α and β are known to be clinically relevant to date. Unstimulated tissue mast cells continuously secrete immature monomeric α and β-protryptase, which constitute the serum basal tryptase. The mature forms of tryptase are only released following mast cell activation [2].

While the exact function of enzymatically inactive protryptase (stored in the endoplasmic reticulum) is still unknown, many studies have highlighted the biological activities of tryptase tetramers. These activities include roles in the acute phase of MC degranulation and in the late phase of inflammation [2,3].

### 3.1. Tryptase Measurement

In clinical practice, measuring total serum tryptase levels as an MC biomarker is a simple and affordable method. This is typically performed by a commercial in vitro quantitative test using a noncompetitive two-site fluorescent enzyme immunoassay (Phadia, ImmunoCAP/Portage, MI, USA). The units used are micrograms per liter (µg/L) or the equivalent nanogram per milliliter (ng/mL) [4,5]. Serum and plasma are the recommended samples for measuring levels of tryptase. In some cases, other bodily fluids, such as tears, saliva, nasal secretions, sputum or feces can be used as well. Urine is not suitable for the analysis, as tryptase is not excreted through the urinary tract [6]. False-positive results due to interference with the rheumatoid factor have been reported and yet overcome by adding heterophilic antibody suppressors and the F(ab’)2 detection antibody [7].

### 3.2. Serum Baseline Tryptase: Normal Range, Determinants and Clinical Implications

There is an important difference between sBT, which represents the total body MC burden, and the transient peak tryptase or sAT level following the MC degranulation [6].

Since the introduction of the total tryptase assay, which has been used worldwide since 1994, the cut-off value or the 95th percentile value for sBT has repeatedly been going down from the initial 13.5 µg/L (1994) to the current 8.4 µg/L (2022). This change can be explained by the major discovery of an autosomal dominant genetic disorder called hereditary alpha-tryptasemia (HαT) and the improved detection of other MC-related conditions [6].

Based on the working classification, the predisposing factors are divided into major (strong), which cause an increase in sBT, and minor (weak) determinants, which cause sBT variation within the reference range [6].

Inherited conditions, such as hereditary HαT, are reported as major determinants. This was confirmed by Waters et al., since almost two-thirds of the fifty-eight patients enrolled in their study with sBT levels of 11.5 μg/L and greater have been diagnosed with HαT [8]. HαT is an autosomal dominant genetic trait with variable clinical penetrance that is caused by an excess of copies of the TPSAB1 gene that encodes α tryptase. This condition exhibits a gene dosage effect with an estimated rise of approximately 9 ng/mL in sBT levels for every additional copy of the TPSAB1 gene [9]. Patients with this genetic trait have levels of sBT above 8 µg/mL and have a 3- to 4-fold increased risk of developing severe anaphylaxis, as reported by various studies [10,11]. Meanwhile, the majority of patients with systemic mastocytosis carry the somatic D816V mutation in the KIT gene [12,13,14,15].

The other major determinants include other MC conditions such as mastocytosis, myeloid disorders, atherosclerosis, acute coronary syndrome, chronic kidney disease (CKD) and gastrointestinal disorders, such as IBS, inflammatory bowel disease, eosinophilic esophagitis, hepatocellular carcinoma and parasitic infections (details listed in Table 1) [15,16,17,18,19,20,21,22,23,24,25,26,27,28]. The minor determinants are not as well defined and include age, sex and lifestyle factors. Therefore, an sBT level above 8.4 µg/mL should elicit further investigation [15].

For patients with a measured sBT greater than 11.5 ng/mL, a clinical algorithm recommended by Waters et al. includes conducting a complete blood count with a leukocyte count, creatinine and estimated glomerular filtration rate measurement, TPSAB1 genotyping, 24 h urinary methylhistamine detection and a KIT p.D816V allele-specific PCR of a blood sample [8,14,15]. All patients with persistent elevation of sBT above 20 ng/mL must be referred to a hematologist for further examination [15].

On the other hand, diagnosing conditions such as anaphylaxis requires correlating the serum acute tryptase (sAT) level with the measured baseline tryptase (sBT) level. Since MCs are located in peripheral tissue and not the bloodstream, there is a delay of 20 to 30 min from the onset of symptoms to the detection of tryptase in the blood, with the levels peaking approximately 1 h after MC activation and symptom onset. To support the clinical diagnosis of anaphylaxis, at least two serum tryptase measurements need to be taken at the appropriate time (Figure 1) [6,29].

Interestingly, the measurement of the tryptase level can also be used in forensic medicine as a promising diagnostic biomarker of sudden death caused by anaphylactic shock, notably when levels of post-mortem tryptase are greater than 30.4 μg/L [30,31].

## 4. The Role of Tryptase in Allergic Diseases

### 4.1. Type I Hypersensitivity Reactions

According to the latest World Allergy Organization (WAO) definition, anaphylaxis is a severe, life-threatening systemic hypersensitivity reaction characterized by a rapid onset with potentially life-threatening airway, breathing or circulatory issues, typically accompanied by skin and mucosal changes [32].

The diagnostic criteria for anaphylaxis proposed by the European Academy of Allergy and Clinical Immunology (EAACI) include the acute onset of the allergic reaction following the immediate exposure to a known allergen involving the skin and mucosal tissue, both in combination with respiratory and cardiovascular compromising, hypotension or a sudden drop in blood pressure (BP) as the sole criteria [33].

The EAACI task force suggests measuring serum tryptase levels 30 min to 2 h after the onset of symptoms, as well as obtaining sBT at least 24 h after the complete symptom resolution, to support the diagnosis of anaphylaxis [34,35]. These timings reflect the half-life of tryptase, which is approximately 2 h, with levels peaking 1 to 2 h after the onset and usually returning to baseline within 6 to 8 h. If a prior sBT or a blood sample collected before the anaphylaxis reaction is available, it can also be used to determine the change in tryptase levels. There is a widely used formula, known as the 20 + 2 rule, recommended for calculating a personalized cut-off value: 1.2 × sBT = 2 [36]. A rise in sAT levels greater than the value defined by the equation had a high specificity and positive predictive value, with a moderate negative predictive value and sensitivity (91%, 98%, 44% and 78%, respectively) [15].

In 2022, Allyson et al. introduced a new model using an sAT/sBT ratio of 1.685, which maximizes both specificity and sensitivity for confirming anaphylaxis in all patients. Their study demonstrated that an sAT/sBT ratio of 1.685 or higher is positively correlated with the clinical diagnosis of anaphylaxis, achieving 95% sensitivity and specificity [15].

It is important to emphasize that serum tryptase is not always elevated in anaphylaxis, especially in food triggers across all ages. Therefore, failing to find an elevated tryptase level does not exclude anaphylaxis [33].

In real-life clinical scenarios, several factors influence the process, including the rapidity of clinical development, the time between the onset of symptoms and the intervention by the emergency department or medical team, as well as the time it takes to stabilize the patient before additional diagnostic tests can be performed. All this affects the timing of the acute serum tryptase sampling, but it should not be disregarded. Previous studies have shown the improved sensitivity of sAT upon serial measurement. The high specificity of sAT makes it a valuable test to distinguish anaphylaxis from its “mimics,” even if it only serves to confirm a past event [37].

#### 4.1.1. Tryptase Values in Insect Venom Allergy

Hymenoptera venom allergy (HVA) is a form of IgE-mediated degranulation of mast cell and basophils [38,39]. Clinical manifestations of HVA can range from large local reactions (LLRs) at the sting site, characterized by localized swelling lasting more than 24 h and exceeding a diameter of 10 cm, to systemic sting reactions (SSRs). SSRs can vary from moderate reactions such as generalized urticaria, angioedema, dizziness, dyspnea and nausea to severe reactions such as shock, loss of consciousness and respiratory or cardiac arrest [40]. The diagnosis of HVA is based on a detailed clinical history and relevant venom by in vivo or in vitro testing [41]. It is recommended to measure the levels of sBT in all patients with a positive history of SSR to identify those at a higher risk of developing severe reactions. Patients can be divided into groups based on their risk of developing SSR using their sBT levels: the low-risk group, with tryptase levels lower than 4 μg/L; the intermediate-risk group, with levels ranging from 4 to 7.5 ng/mL and the high-risk group with levels greater than 7.5 μg/L [42]. A measured value greater than 11.4 µg/L often indicates underlying mastocytosis, which is more frequent in HVA patients than in the general population and is shown to be an independent risk factor for more severe SSR [43]. Therefore, EAACI guidelines recommend prolonged venom immunotherapy (VIT) in all patients with mastocytosis and/or sBT higher than 11.4 µg/L because of a higher risk of treatment failure and a higher chance of relapse after stopping VIT, which includes fatal reactions. It still remains unclear whether VIT should be lifelong for those patients [44].

#### 4.1.2. Drug Allergies and Tryptase

Drug hypersensitivity reactions (DHRs) can be classified as either immediate or delayed, based on their onset time. Immediate DHR is usually IgE-mediated and occurs within 1–6 h after drug administration, following the binding to the appropriate cell receptor that stimulates the release of various mast cell mediators, including tryptase. DHR can also be non-IgE-mediated, leading to direct mast cell degranulation through a different pathway. Clinical manifestations of immediate DHR include urticaria, angioedema, conjunctivitis, bronchospasm or even anaphylaxis [37].

Anaphylaxis usually occurs suddenly and with severe symptoms, requiring prompt recognition and treatment. Considering that different pathological mechanisms leading to drug-induced anaphylaxis can be similar, the presentation alone cannot solemnly confirm the type of reaction. Therefore, measuring tryptase levels is considered to be a helpful tool for timely diagnosis. These findings are particularly important for patients who develop perioperative anaphylaxis [45].

#### 4.1.3. Use of Tryptase in Perioperative Anaphylaxis

Perioperative anaphylaxis (PA) is a form of systemic anaphylactic reaction caused by antibiotics, neuromuscular blocking agents (NMBAs), chlorhexidine or latex. It is frequently unrecognized due to various factors: a personal history of diseases, lack of adequate symptom description due to a sedated or anesthetized patient, similar pharmacodynamic characteristics of anesthetics and anaphylaxis symptoms. These factors make PA a medical emergency which is very difficult to treat [45].

The estimated incidence of PA varies between 1:6000 and 1:20,000 cases, with a mortality rate of up to 9% [21]. Many studies have identified NMBA as the most common trigger, while others state that antibiotics are the most frequent cause [33,45].

Mast cell-derived tryptase measurement can be helpful in confirming the diagnosis of PA, although the time it takes to run this test is a limiting factor for immediate diagnosis [15]. According to EAACI recommendations, measuring sAT (0.5–2 h after the initiation of the symptoms) and sBT (before onset or at least 24 h after resolution of the symptoms) levels increases the diagnostic accuracy of PA, as well as the number of referrals to an allergist for a further workup [34].

##### Clinical Case No. 1

A female patient was admitted to the vascular surgery department for brachycephalic truncus reconstruction and was administered Midazolam, Propofol, Fentanyl, Rocuronium, Sevoflurane and Cefuroxime. Almost immediately, she experienced bronchospasm, followed by hemodynamic instability and resistant hypotension. The patient was treated with adrenalin, corticosteroids, antihistamines and bronchodilators, which resulted in a favorable therapeutic response. Immunological analyses revealed an elevated sAT level of 18.9 µg/L (reference range is 0–11 mcg/L) compared to an sBT level that was 5.69 µg/L, confirming the assumed diagnosis of PA. The specific IgE testing for cephalosporins was negative. Considering the patient’s history of previous surgical procedures under general anesthesia, which could have caused the initial sensitization, in vivo drug provocation tests with selected alternative intravenous anesthetics were performed. The tests for Remifentanyl and Hypnomidate were negative. However, a positive intradermal test with Midarine lead to the conclusion that the NMBAs were the most probable cause of the PA. Therefore, it was concluded that NMBAs should be avoided or replaced in future procedures.

#### 4.1.4. Tryptase and Iodinated Contrast Media Hypersensitivity Reactions

Contrast media agents are a rare cause of potentially severe anaphylactic reactions. Due to their wide use and broad spectrum of clinical indications, iodinated contrast media agents (ICMs) are the most common culprits. The type of contrast media agent and its osmolarity and the dosage and rapidness of the application play an important role when it comes to both allergic and non-IgE-mediated reactions that can be both responsible for the MC degranulation and serum tryptase level rise. However, there are many contributing factors such as advanced age and comorbidities (allergic and cardiovascular diseases) that may predispose a certain individual to an unfavorable outcome.

##### Clinical Case No. 2

A female patient undergoing a computed tomography scan (CT) of the abdomen experienced a severe reaction during the administration of the ICM. She lost consciousness, developed cyanosis with agonal breathing, had non-palpable peripheral pulses and her blood pressure was immeasurable. Immediately, she was intubated alongside the continued resuscitation measures of electro cardioversion, crystalloid infusion and vasopressor support, including adrenalin. Due to her clinical presentation of generalized erythema and periorbital edema, she was treated with corticosteroids and antihistamines. The diagnosis of anaphylaxis was confirmed by serum tryptase analysis (sAT level was 160 µg/L and sBT level 11 µg/L). The patient had an atopic constitution and a previous medical history of allergic rhinitis and asthma, as well as nasal polyposis with sensitization to house dust mites and tree pollen. Despite having no prior drug hypersensitivity reactions and having undergone CT pulmonary angiography a year earlier without incident, premedication was not administered. The patient spent two weeks in the intensive care unit but succumbed to a severe brain injury. This case emphasizes the importance of detailed overall clinical and risk assessments, critical patient selection, timely premedication and therapeutic intervention.

## 5. Conclusions

Based on the thorough literature research and the clinical experience, illustrated by two cases from our own clinical practice, we have arrived at the following conclusions:-In all suspected cases of MC activation and degranulation, a timely tryptase measurement is required to confirm anaphylaxis and differentiate it from its mimics.-The laboratory analysis is time-consuming but should not be neglected, nor should the initial treatment of the acute reaction be postponed while waiting for the said result.-Tryptase levels determined upon suspected drug allergy may help with risk stratification and the need for premedication during imaging procedures and surgical interventions.-An elevated sBT level in HVA is associated with a higher risk of anaphylactic reactions and therefore requires a prolonged VIT.-The constant increase in the number of clinical conditions associated with a noticeable elevation of the sBT level requires a multidisciplinary approach and collaboration, as well as further scientific research and the continuation of medical education.

## Figures and Tables

**Figure 1 medicina-60-01380-f001:**
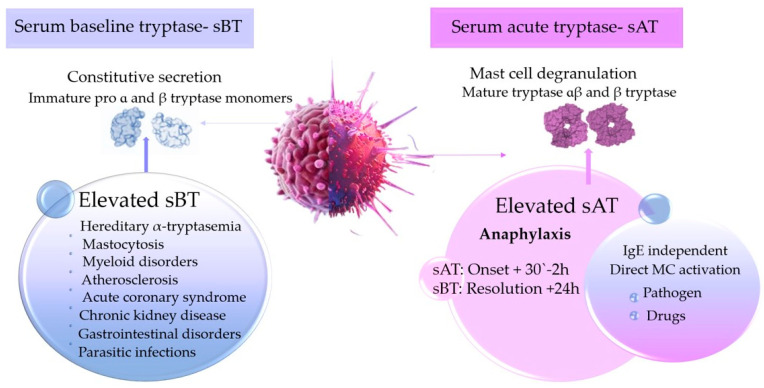
Clinical significance of serum tryptase measurement in acute reactions and chronic conditions.

**Table 1 medicina-60-01380-t001:** Role of elevated tryptase level in various clinical implications.

Clinical Implications		Role of Elevated Tryptase Level
Allergic diseases		
➢ Type I hypersensitivity reactions	Anaphylaxis	Confirmation of the diagnosis of severe hypersensitivity reaction
➢ Drug allergy	Perioperative anaphylaxsis ◆ Antibiotics ◆ Neuromuscular blocking agents (NMBA) ◆ Chlorhexidine ◆ Latex Iodonated contrast media hypersensitivity reactions	Confirmation of the mast cell degranulation process regardless of the pathophysiological mechanism Both IgE- and non-IgE-mediated reactions
➢ Insect venom allergy	Hymenoptera venom allergy	Confirmation of the diagnosis of anaphylactic reaction to Hymenoptera Risk stratification Higher risk of treatment failure and a higher chance of relapse after stopping venom immunotherapy which includes fatal reactions Treatment management Prolonged venom immunotherapy in all patients with mastocytosis and/or sBT > 11.4 µg/L
Non-allergic conditions		
➢ Mast cell-related disorders	◆ Primary clonal disorders (mastocytosis) ◆ Secondary causes of mast cell activation ◆ Idiopathic	Disease confirmation All patients with sBT > 20 µg/L should be referred to a hematologist for further diagnostic examination
➢ Renal disorders	Chronic kidney disease	Prognostic biomarker Higher levels of sBT correlate with a greater renal impairment, creatinine and proteinuria
➢ Hematological malignancies	Acute myeloid leukemia Chronic myeloid leukemia Chronic eosinophilic leukemia	Diagnostic and prognostic biomarker
➢ Cardiovascular diseases	Atherosclerosis Acute coronary syndrome	Biomarker of advanced peripheral atherosclerosis and development of aneurysms Possible long-term prognostic biomarker
➢ Gastrointestinal diseases	Irritable bowel syndrome	Confirmation of the diagnosis Higher sBT levels in IBS-D patients
	Eosinophilic esophagitis	Significantly increased serum tryptase levels suggest diffuse diseases like eosinophilic gastroenteritis and extraesophageal symptoms
	Inflammatory bowel disease	Biomarker of fibrosis Indicator of the extra intestinal manifestations
	Malignancies Hepatocellular carcinoma	Correlation with tumor angiogenesis Monitoring the response to treatment
➢ Parasitic infections	*Trypanosoma species*, *Toxoplasma gondii*, *Plasmodium species*, *Leishmania species*, *Onchocerciasis*	Diagnostic and monitoring biomarker

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
