# Peer review of "Tryptase: The Silent Witness of Past and Ongoing Systemic Events"

_medicina, 2024, doi:10.3390/medicina60091380_

Round 1

Reviewer 1 Report

Comments and Suggestions for Authors

Dear Authors,

Your manuscript has several fundamental issues that need to be addressed to meet the publication standards. Please consider the following points:

1. **Abstract:** The abstract should include the objective and the overall result so that the reader can quickly grasp the contribution of the paper.

2. **Repetitive Sentences:** Sentences are exactly repeated in the text and need to be revised. For example, the first paragraph of the introduction is exactly repeated in the abstract.

3. **Unclear Sentences:** Some sentences are unclear and ambiguous. For example, the sentence in line 49 needs to be clarified.

4. **Introduction Format:** The format of the introduction is not appropriate. The last paragraph of the introduction should mention the innovation and the objective of the project.

5. **Content Organization:** The organization of the content is not suitable, and some headings are discussed in a miscellaneous manner. The topics should be properly categorized, and the use of tables can enhance the clarity and attractiveness of the paper.

Author Response

Thank You for the time dedicated to thorough review of this manuscript. Detailed responses are stated below, alongside the corrections that are marked in red in order to track the changes in the re-submitted file.

Comment 1: **Abstract:** The abstract should include the objective and the overall result so that the reader can quickly grasp the contribution of the paper.

Response 1: Thank You for bringing this to our attention. Therefore, we have made significant changes to the abstract: marking the separate sections of the article. In the re-submitted form, both the objective and conclusions are stated out. We are hoping that said changes will make the article more comprehensible and emphasize the importance of the serum tryptase in clinical practice.

Comment 2:  **Repetitive Sentences:** Sentences are exactly repeated in the text and need to be revised. For example, the first paragraph of the introduction is exactly repeated in the abstract.

Response 2: We agree with this remark and have, accordingly revised and modified the text, starting from the abstract and the introduction in order to keep the same point of the text, without blunt repetition.

Comment 3: **Unclear Sentences:** Some sentences are unclear and ambiguous. For example, the sentence in line 49 needs to be clarified.

Response 3: Thank You for the comment. The sentence from the previously stated line 49 (now line 63-69, paragraph referring to the mast cell-derived tryptase) has been changed in order to make this paper easier to read and understand. The slight technical changes have also been made to the paragraph 4.5. referring to the role of tryptase in gastroenterology. 

Comment 4. **Introduction Format:** The format of the introduction is not appropriate. The last paragraph of the introduction should mention the innovation and the objective of the project.

Response 4: Thank You for pointing this out. The introduction has been modified and the objective included in the re-submitted form, casting a light into novel findings regarding the potential roles of tryptase.

Comment 5. **Content Organization:** The organization of the content is not suitable, and some headings are discussed in a miscellaneous manner. The topics should be properly categorized, and the use of tables can enhance the clarity and attractiveness of the paper.

Response 5: Thank You sincerely for providing us with that information. We completely agree so the article has also been restructured, according to the reviewer remarks. The organization of the content was modified, with gradual development from the basic biological and biochemical characteristics of the tryptase, to molecular and further clinical implications. The paper now addresses the traditional role of tryptase in allergic diseases and other/ mast cell related disorders. The last section is dedicated to novel roles and wider clinical spectrum that aims to highlight the full potential as a novel and crucial biomarker. All of the changes are marked red in the re-submitted text. We hope that the incorporated table, formulated upon Your suggestion will provide a quick grasp of the content of this paper, enhancing its clarity while bringing the reader’s attention to the numerous roles and benefits of tryptase acquisition and use in everyday clinical practice. 

Reviewer 2 Report

Comments and Suggestions for Authors

In this review manuscript, entitled “Tryptase- The silent whiteness of the past and ongoing systemic events,” the authors evaluate the importance of tryptase as a biomarker in mast cell-related conditions and severe hypersensitivity reactions. This manuscript provides vast information on the timeframe where the secretion of serum tryptase could be detected, the range of tryptase levels in various pathological conditions, and the use of tryptase in perioperative anaphylaxis supported by two clinical cases. The manuscript is well-written, and the references are well-cited. I do not have any major critics except for a few minor corrections required:

1.     Line 157 “et all 2022”

2.     Line240 “levelr ise”

Comments on the Quality of English Language

 English is fine, but a spell check is required.

Author Response

Thank You for the time dedicated to thorough review of this manuscript. Detailed responses are stated below, alongside the corrections that are marked in red in order to track the changes in the re-submitted file.

Regarding the review: In this review manuscript, entitled “Tryptase- The silent witness of the past and ongoing systemic events,” the authors evaluate the importance of tryptase as a biomarker in mast cell-related conditions and severe hypersensitivity reactions. This manuscript provides vast information on the timeframe where the secretion of serum tryptase could be detected, the range of tryptase levels in various pathological conditions, and the use of tryptase in perioperative anaphylaxis supported by two clinical cases. The manuscript is well-written, and the references are well-cited. I do not have any major critics except for a few minor corrections required:

Comment 1:   Line 157 “et all 2022”

Response 1: Thank You for the kind words and for pointing this out. The change in the line (previously 157, now 175) has been made.

Comment 2:     Line240 “levelr ise”

Response 2: Thank You for bringing this to our attention. The said error in the line previously 240 (now 258) has been corrected.

Comment 3: Comments on the Quality of English Language: English is fine, but a spell check is required.

Response 3: Thank You for the suggestion. The spell check has been done, and all of the changes made to this article have been marked in red in the re-submitted article.

Reviewer 3 Report

Comments and Suggestions for Authors

The paper presents valuable description of the use of tryptase as important biomarker in mastocite related condition including anaphylaxis.

In order to improve the quality of the paper I have following recommendations:

1. There are many technical errors through out the text- spaces are missing and citation of references should be before the end of the sentence, eg: instead of  ....  sensitivity reactions. (1,2) it should be ....  sensitivity reactions (1,2).

Line 71: in vivo should be in italic;

Line 157: Allyson et al in 2022.   space before 2022

Line 180: within 1-6h, space before 1-6h

Passus 2.1.1 is not clearly written. Is it the short cat study of female patient? Term she is usually used in this pasus. Please reformulate this section and be more clear what was your point here to describe.

Line 401. Trypanosome (please write in italic)

T. gondii should be T. gondii

Plasmodium (in italic)

Leishmania should be Leishmania italic.

Author Response

Thank You for the time dedicated to thorough review of this manuscript. Detailed responses are stated below, alongside the corrections that are marked in red in order to track the changes in the re-submitted file.

Comment 1. The paper presents valuable description of the use of tryptase as important biomarker in mastocyte related condition including anaphylaxis. In order to improve the quality of the paper I have following recommendations:

There are many technical errors throughout the text- spaces are missing and citation of references should be before the end of the sentence, eg: instead of  ....  sensitivity reactions. (1,2) it should be ....  sensitivity reactions (1,2).

Response 1: Thank You for the kind words and for pointing this out. The technical errors throughout the text have been corrected, including the spacing and form of citation.

Comment 2: Line 71: in vivo should be in italic;

Response 2: Thank You for the provided information. The said error in the line 71 previously (now 86) has been corrected.

Comment 3: Line 157: Allyson et al in 2022.   space before 2022

Response 3: According to the suggestion, the change in the line (previously 157, now 175) has been made.

Comment 4: Line 180: within 1-6h, space before 1-6h

Response 4: Thank You for the stated technical error, the change in the line 180 from the previous version of the text (now 199) has been modified accordingly.

Comment 5: Passus 2.1.1 is not clearly written. Is it the short cat study of female patient? Term she is usually used in this passus. Please reformulate this section and be more clear what was your point here to describe.

Response 1: Thank You for bringing this to our attention. The section is now labeled 3.1.1, due to the overall paper reconstruction, according to the received revisions. The clinical cases have been modified, in order to transmit a clear message of the clinical implications regarding the timely tryptase acquisition in the setting of perioperative anaphylaxis and suspected drug allergy as a part of the proposed clinical algorithm.       

Comment 6: Line 401. Trypanosome (please write in italic)

  1. gondii should be T. gondii

Plasmodium (in italic)

Leishmania should be Leishmania italic

Response 6: We kindly thank You for the provided technical information. All of the stated names of the causes of the parasitic infection have been written in italic- the changes made have been noted in the re-submitted text in red marking (lines 420, 421 and 423).